# Integrating Human Mobility Models with Epidemic Modeling: A Framework for Generating Synthetic Temporal Contact Networks

**DOI:** 10.3390/e27050507

**Published:** 2025-05-08

**Authors:** Diaoulé Diallo, Jurij Schoenfeld, René Schmieding, Sascha Korf, Martin J. Kühn, Tobias Hecking

**Affiliations:** 1Institute of Software Technology, German Aerospace Center (DLR), 51147 Cologne, Germany; jurij.schoenfeld@dlr.de (J.S.); rene.schmieding@dlr.de (R.S.); sascha.korf@dlr.de (S.K.); martin.kuehn@dlr.de (M.J.K.); tobias.hecking@dlr.de (T.H.); 2Life and Medical Sciences Institute and Bonn Center for Mathematical Life Sciences, University of Bonn, 53127 Bonn, Germany

**Keywords:** temporal contact networks, epidemic simulation, human mobility models, synthetic network generation, agent-based modeling

## Abstract

High-resolution temporal contact networks are useful ingredients for realistic epidemic simulations. Existing solutions typically rely either on empirical studies that capture fine-grained interactions via Bluetooth or wearable sensors in confined settings or on large-scale simulation frameworks that model entire populations using generalized assumptions. However, for most realistic modeling of epidemic spread and the evaluation of countermeasures, there is a critical need for highly resolved, temporal contact networks that encompass multiple venues without sacrificing the intricate dynamics of real-world contacts. This paper presents an integrated approach for generating such networks by coupling Bayesian-optimized human mobility models (HuMMs) with a state-of-the-art epidemic simulation framework. Our primary contributions are twofold: First, we embed empirically calibrated HuMMs into an epidemic simulation environment to create a parameterizable, adaptive engine for producing synthetic, high-resolution, population-wide temporal contact network data. Second, we demonstrate through empirical evaluations that our generated networks exhibit realistic interaction structures and infection dynamics. In particular, our experiments reveal that while variations in population size do not affect the underlying network properties—a crucial feature for scalability—altering location capacities naturally influences local connectivity and epidemic outcomes. Additionally, sub-graph analyses confirm that different venue types display distinct network characteristics consistent with their real-world contact patterns. Overall, this integrated framework provides a scalable and empirically grounded method for epidemic simulation, offering a powerful tool for generating and simulating contact networks.

## 1. Introduction

The modeling of infectious disease spread is a critical component in understanding and addressing the impact of pandemics. Accurate simulations can help inform public health policies, guide intervention strategies, and contribute to preparedness for future outbreaks. The utility of such simulations often relies on the quality of the underlying data, which can range from averaged demographic data obtained through census studies (see, e.g., [1]) to high-resolution temporal contact patterns that capture individual interactions in detail (see, e.g., [2]).

Contact networks provide an accurate and flexible way to represent these human interactions [3]. By modeling individuals as nodes and their interactions as edges, they create a framework for the structural analysis of how infections spread through a population. Depending on the level of detail required, the temporal resolution of these networks can vary widely. Networks aggregating contact patterns over longer periods, such as days, weeks, or even months, are especially useful for analyzing general trends and understanding population-level dynamics, as highlighted in [4,5].

Various methods for gathering real-world mobility and contact data exist, each with specific challenges. Cellular signaling data (CSD), for example, can capture large-scale movement patterns through mobile phone records, as demonstrated during the COVID-19 pandemic in China [6]. While such aggregated data enable valuable insights into population-level mobility dynamics, they raise substantial privacy concerns [7]. Moreover, CSD typically provides coarse spatial and temporal resolution compared to direct interaction measurements, posing additional challenges for fine-grained epidemic modeling.

In contrast, high-resolution contact networks focus on shorter timescales and finer spatial resolution, capturing interactions in intervals as short as hours or even minutes and within specific physical locations. These networks are especially valuable for studying individual risks of infection. For example, high-resolution data enable researchers to explore digital contact tracing strategies, which play an important role in the containment of infections [8,9,10]. This includes providing risk-based recommendations as well as assessing the effectiveness of interventions.

Although high-resolution contact network data are crucial, gathering such data come with substantial challenges. These types of data need to be both temporally detailed and highly accurate, reflecting the precise timing and duration of interactions. Yet, most existing empirical temporal networks are limited to specific settings—like schools, universities, or supermarkets—and do not capture interactions across varied environments. This limitation stems from the inherent challenges of recording human interactions. Typically, empirical studies in this domain gather contact data through wearable sensors designed to track face-to-face interactions [11,12,13,14,15]. While this enables the collection of contacts in a specific environment for a set of individuals, this method cannot account for larger populations or multiple environments.

Epidemic simulation frameworks, such as Covasim [16], FRED [17], OpenCOVID [18], GEMS [19], Repast [20,21], and MEmilio [22], to only name a few, use various models, including compartmental and agent-based models, to study the dynamics of infectious diseases. However, if these frameworks do not use simplified contact patterns, such as uniform mixing or random networks, which reflect real-world human interactions only on an aggregated level, they are in need of complex contact information to be initialized.

Building on our previous work [23,24] that optimized microscopic human mobility model (HuMM) parameters via Bayesian optimization—ensuring that the generated temporal contact networks mirror empirical reference networks—we integrate this approach into the agent-based model [25] of the MEmilio simulation framework [22]. By incorporating HuMMs [26,27] to model encounters in various location types such as schools, workplaces, supermarkets, and public spaces, the extended MEmilio framework produces synthetic, high-resolution contact networks that more realistically reflect population interactions than homogeneous mixing. This integrated approach provides a scalable method for simulating temporal contact networks. We provide an extensive analysis of the resulting temporal networks, emphasizing both their realistic structural characteristics and their potential utility for epidemic research. By integrating HuMMs into MEmilio and generating realistic temporal contact networks, our work contributes to the development of more accurate epidemic simulations and enables more precise evaluation of intervention techniques.

The remainder of this paper is organized as follows. Section 2 reviews the state of research on temporal contact networks and epidemic simulation frameworks, highlighting the limitations of existing approaches. Section 3 introduces our integrated MEmilio–HuMM framework, detailing how Bayesian-optimized human mobility models are used to generate realistic, high-resolution contact networks. Section 4 explains the specific procedures for contact network generation, including the modeling of households and diverse location capacities. Section 5 presents a comprehensive analysis of the resulting temporal contact networks, examining both their structural properties and epidemic behavior under varying population sizes and location capacities. Finally, Section 6 concludes the paper by summarizing the key findings and discussing potential directions for future research.

## 2. Related Work

### 2.1. Temporal Contact Networks

High-resolution temporal contact data are critical for understanding person-to-person transmission dynamics. Several notable empirical datasets capture such interactions in restricted settings; for example, the Sociopatterns high school [15] and primary school [11] networks, or the office network from Génois and Barrat [12]. In these studies, wearable proximity sensors recorded interactions at short intervals (often seconds), providing invaluable insights into contact structure and duration. However, most of these datasets are limited in both scale (a few hundred individuals) and scope (single environments such as a specific school or office). Although these empirical sources offer a granular view of contact formation, they are limited to selected and particular settings. Realistic large-scale epidemic scenarios often require mobility and interaction data across numerous locations (e.g., households, schools, workplaces, and public spaces). In practice, complete coverage at such a scale is nearly impossible to collect. This gap motivates synthetic approaches that can generate rich, time-resolved networks on large agent populations while preserving empirically validated characteristics.

### 2.2. Epidemic Simulation Frameworks

Numerous models and modeling platforms have emerged over the last decades to simulate epidemic spread over synthetic or semi-synthetic contact networks—see, e.g., [1,16,17,18,19,20,21,22,25,28,29,30,31,32,33,34,35,36,37,38], to name only a few.

Covasim [16], for instance, uses contact network-based modeling, incorporating data-driven synthetic population networks or hybrid approaches requiring less data. While Covasim provides flexible tools for individual-level risk analysis, it does not incorporate explicit location-specific human mobility.

OpenCOVID [18] is another individual-based simulator using an age-structured population network and accounting for risk groups, seasonal effects, and various non-pharmaceutical interventions (NPIs). Although OpenCOVID incorporates several realistic features, it uses contact patterns largely through aggregated contact rates and does not integrate location-specific high-resolution mobility at scale.

MEmilio [22,25] (the framework extended in this work) supports both differential equation- and agent-based models. By default, its agent-based component assumes largely uniform mixing within each location (e.g., households or workplaces). While this assumption simplifies simulation design, it does not capture realistic micro-scale movement and time-varying contact patterns. Nonetheless, MEmilio’s explicit representation of locations provides a flexible structure for integrating representative network models, enabling more accurate modeling of human mobility within locations.

Across these frameworks, an open challenge persists: how to move beyond aggregated or static snapshots to achieve truly dynamic, location-aware contact networks spanning large populations. The present work contributes to this challenge by coupling a state-of-the-art epidemic simulation framework with Bayesian-calibrated mobility models that generate time-resolved, environment-specific contact structures—a crucial step toward realistic modeling of human mobility and more effective evaluation of targeted control strategies.

## 3. Methodology

### 3.1. MEmilio Simulation Framework

The MEmilio [22] framework is an open-source software framework designed to simulate the spread of infectious diseases using a variety of epidemiological models. Initially focused on metapopulation models [1,39], it now offers integro-differential equation-based models [37], agent-based models (ABMs) [25], hybrid metapopulation-agent-based models [38], and (graph) neural network surrogate models [40]. Therefore, the framework provides the possibility to handle coarse-grained models focusing on population-level dynamics as well as simulations of individual-level interactions. Its modular design allows combining different disease dynamics with tailored mobility patterns, making it suitable for modeling a wide range of pathogen transmission scenarios. MEmilio uses an object-oriented design approach and models individuals as *Person*s that can form *Household*s of different sizes, and meeting or venue spaces as *Location*s. Additionally, it uses structures for face *Mask*s of different types, *TestingStrategy*(ie)s which can handle different *TestingScheme*s (e.g., mandatory tests before entering a location), *TripList*s that store individual activity-driven mobility patterns, or *mobility rules* for generic/collective/(non-individual) location changes. The framework is implemented in efficient and templated C++ to allow for fast execution and to use general concepts such as parameter spaces or mid- to long-distance mobility patterns in a graph pattern. To be run in parallel on small clusters as well as on supercomputing infrastructure, essential code parts are parallelized with OpenMP for shared memory patterns and with the MPI standard for ensemble runs on distributed memory patterns. More technical details on, e.g., logging and I/O routines, random number generators, validation efforts, and broader functionalities of MEmilio can be found in its official documentation (available online: https://github.com/SciCompMod/memilio (accessed on 3 April 2025)) and a detailed overview of the ABM can be found in [25]. In the present work, we use MEmilio solely as a generator for synthetic temporal contact networks; all infection-related components are deactivated during our simulations.

Currently, the ABM from MEmilio modified in this work represents locations as fully or randomly connected networks, meaning that every individual at a location is assumed to have contact with every other individual present. Between these locations, agents can move from one location to another. In its most basic form, the model relies on static movement rules that determine each agent’s behavior—for example, whether and when they go to work based on age and internal parameters, or if they visit a hospital when severely ill. While these rules can be expanded and a more sophisticated method based on a traffic model is available, the basic setup already captures general population movement realistically. For more details, see [25].

While variations in transmission probabilities at each location are incorporated using age-specific contact rates derived from given age matrices, this approach relies on aggregated information about human contact patterns. Furthermore, the transmission process does not yet account for the distinct characteristics of different environments, which play a significant role in shaping actual human interaction patterns.

In this paper, we build on MEmilio’s architecture to incorporate Bayesian-optimized human mobility models for generating high-resolution temporal contact networks. Rather than relying on default assumptions about mixing patterns or uniform interaction probabilities, we extend the framework to model realistic micro-scale movement patterns for specific locations. As discussed in the following sections, this approach allows us to capture detailed, time-resolved contact structures within heterogeneous environments such as households, schools, workplaces, and public spaces. Our aim is to enable epidemic simulations that reflect the complex, time-varying nature of human interactions. The integration of Bayesian-optimized human mobility models with MEmilio sets the stage for flexible and parameterizable temporal contact network generation.

### 3.2. Bayesian-Optimized Human Mobility Models

Human Mobility Models (HuMMs) have proven to be an effective way to capture the spatial and temporal nuances of person-to-person encounters in confined environments [24,26,27,41,42]. Compared to naive approaches to human movement patterns, such as random or uniform distributions, HuMMs explicitly describe how individuals move through space, pause, and change their trajectories in a manner that can mirror real-world behaviors. Nevertheless, designing HuMMs that align with empirical observations requires choosing suitable parameter values (e.g., movement velocities, distances traveled, pause durations). Moreover, a single model may have many parameters governing various aspects of the simulated mobility, making a manual search for an optimal configuration impractical. In our prior work [24], we employed Bayesian optimization to tune the parameters of HuMMs, ensuring that the simulated contact networks align with observed infection curves and the network properties of empirical reference networks. Building on this foundation, we integrate this approach into MEmilio to generate high-resolution temporal contact networks tailored to specific environments. This integration enables a more realistic representation of human mobility, moving beyond simplified assumptions and allowing for detailed, time-resolved modeling of contact structures in diverse settings. In the following, we provide a brief recap of our approach; for a detailed description, we refer to our original work.

#### 3.2.1. Bayesian Optimization Strategy

We formulate a multi-objective loss function, $L(G_{m},G_{e})$, that integrates key differences between the modeled network $G_{m}$ and the empirical network $G_{e}$. In our formulation, we quantify the following aspects:$\Delta I_{max}$: the absolute difference in the peak number of infections;$\Delta T_{I_{max}}$: the relative difference in the timing of these peaks;$\Delta N_{E}$: the relative difference in the total number of edges;$\Delta t_{cd}$: the difference in the contact duration distributions (measured via the Kolmogorov–Smirnov statistic).

These individual metrics are combined into the following single loss function:$$L\left(G_{m},G_{e}\right)=5\cdot \Delta I_{max}+3\cdot \Delta T_{I_{max}}+2\cdot \Delta N_{E}+\Delta t_{cd}.$$
Weights were selected to prioritize faithful reproduction of epidemic peaks, $\Delta I_{\max}$, and their timing, $\Delta T_{I_{\max}}$, while still capturing key network structure via $\Delta N_{E}$ and $\Delta t_{cd}$; in preliminary tests, this configuration yielded the best balance between epidemic-curve fidelity and structural realism.

Running multiple simulations, the Bayesian optimization algorithm searches the high-dimensional parameter space of the HuMMs to minimize $L(G_{m},G_{e})$. In each iteration, a candidate set of HuMM parameters is used to generate a temporal contact network by simulating individual movements and recording face-to-face contacts—defined by a maximum contact distance of 1.5 m and a 120-degree field of view (following our previous work [24]). For each candidate network, we evaluate the loss by running multiple Susceptible-Infected-Recovered (SIR) model simulations. We employ a standard SIR model on the generated temporal contact networks, following the Gillespie algorithm [43]. At each simulation time step, every infected node attempts to infect each of its susceptible neighbors with probability β. Infected nodes recover with probability σ per time step, after which they become immune and cannot distribute further.

Through this iterative procedure—generating candidate networks, evaluating their loss, and refining the HuMM parameters—we converge on a configuration that best replicates the empirical infection dynamics and network topology. The target networks are empirical contact networks; specifically, we use temporal networks recorded in experimental studies and published in prior research as references (see Section 2).

#### 3.2.2. Parameter Space

Although each HuMM (e.g., STEPS, STEPS with RWP [42]) may include model-specific parameters, the general types of parameters we optimize are as follows:*Movement shapes*, which controls whether individuals tend to move short distances with occasional long trips, following a power-law distribution, or have more uniform travel patterns.*Pause dynamics*, describing how long individuals tend to remain in a given spot before continuing movement.*Spatial clustering*, indicating whether individuals are likely to cluster in default sub-spaces and how strongly they gravitate back to these spaces.

The Bayesian optimizer sequentially selects candidate parameter sets, simulates the corresponding contact network using HuMMs, and then runs SIR simulations to assess the network’s performance. In each iteration, it uses the loss function derived from these simulations to update a probabilistic model of the parameter space, directing subsequent parameter searches towards more promising regions. For more details on parameter ranges and the optimization procedure, we refer to our prior work [24].

#### 3.2.3. Stochastic Repetitions and Final Selection

Both the HuMM generation process and the SIR simulations are inherently stochastic. To mitigate the variability introduced by randomness, we generate multiple independent network realizations for each parameter set and run multiple SIR simulations on each network. We aggregate the results to produce a representative performance score. We then select the best parameter set and generate a final ensemble of networks for downstream use.

This process yields diverse contact networks by starting each realization from a different random seed, affecting initial positions, step lengths, and pause times. The optimizer explicitly minimizes the mean loss across this ensemble, promoting parameter sets that perform robustly across a range of plausible contact scenarios. In practice, this stochastic sampling mitigates potential overfitting to the particular structure of a single empirical reference network. Nonetheless, the more empirically distinct reference datasets available for calibration—e.g., contact networks from multiple offices of varying sizes or cultural contexts—the stronger the resulting generalizability of the model.

In the subsequent sections, we illustrate how optimized HuMMs are combined with MEmilio to create realistic high-resolution temporal contact networks covering not only a single location but multiple environments.

### 3.3. Integration of Bayesian-Optimized HuMMs into MEmilio

A key objective of this work is to embed the Bayesian-optimized HuMMs (described in Section 3.2) into the MEmilio simulation framework, thus enabling the generation of realistic temporal contact networks. This integration replaces MEmilio’s default assumption that locations are fully connected (i.e., each agent interacts with all others in the same location). Figure 1 depicts the flow of integration from empirical reference networks to final time-resolved contact networks.

Concretely, we use the empirically derived and Bayesian-calibrated HuMM parameters to generate synthetic, location-specific contact networks for different location types (schools, workplaces, supermarkets, social events) and capacities (how many agents occupy that location). Following our previous work [24], we used existing empirical networks to calibrate our HuMMs (see Section 4.2). These networks capture 24-h spatiotemporal contact patterns, aggregated at hourly resolution.

Once these HuMM-based contact networks are generated and stored in a network library, MEmilio can assign one of these pre-generated networks to each location, according to that location’s type and capacity. As the simulation runs, each agent is mapped onto a node in the location’s adjacency matrix; thus the *fully connected* assumption is replaced by an *empirically grounded*, time-varying network of interactions. Furthermore, we aggregate the outputs of the Bayesian-calibrated HuMMs into a temporal network with an hourly resolution; the framework can be readily adapted to finer temporal scales (e.g., minute-level resolution) if needed. This aggregation step naturally produces edge weights that capture the intensity (i.e., duration) of contacts within each time window, allowing subsequent epidemic simulations to incorporate these weights for more nuanced modeling of transmission dynamics.

We adopt an hourly time resolution to strike a balance between behavioral realism and computational efficiency. Hourly aggregation is fine-grained enough to capture key within-day dynamics relevant for infection transmission—such as morning arrival peaks or afternoon dispersal—while limiting the stochastic noise that can dominate minute-level data. Moreover, for the epidemiological focus of this study, hourly resolution provides a stable and interpretable timescale. While future work could explore higher-resolution variants, we consider the current granularity appropriate for modeling transmission-relevant contact dynamics across large populations and locations. Finer temporal resolutions may capture rapid contact fluctuations more precisely, but they can also increase noise and computational demands; conversely, coarser aggregations risk obscuring critical short-term contact patterns.

In this way, the Bayesian-optimized HuMMs provide individual movements for specific locations, yielding rich network topologies that better reflect real-world data than uniform interaction models. Section 4 details the parameter settings, selected empirical networks, and overall simulation setup. As mentioned in Section 3, epidemic spread is disabled during MEmilio runs to prevent infections from altering the active agent set (e.g., through isolation or death); in the context of this work, the framework serves only as a network generator.

Figure 2 shows the schematic overview of our proposed framework. It illustrates the different types of locations in the simulation—households, schools, workplaces, supermarkets, and social events—each of which can appear in multiple instances. Every instance is associated with a pre-generated, time-resolved contact network (indicated by the layers of networks and the black time arrow), ensuring realistic interaction patterns within that environment. Dashed arrows represent the flow of agents between locations, which is governed by MEmilio’s movement rules.

## 4. Contact Network Generation

This section outlines the key procedures used to generate the temporal contact networks (TCNs) analyzed in our study. Building on the methodology described in Section 3.3, we detail how demographics (household composition) and location capacities are defined, and how we set up different TCN scenarios to investigate the influence of location size and population scale on epidemic dynamics.

### 4.1. Household Composition and Demographic Setup

We employ a representative household composition derived from German census data for 2023 [44]. As shown in Table 1, 41% of households are single-person, 33.5% consist of two persons, 12% have three persons, 9.5% have four persons, and 4% have five or more persons. For simplicity, we assume all households with five or more members consist of exactly five persons. In our model, the population is partitioned into four age groups (0–4, 5–18, 19–64, and 65+). As the census data suggest that roughly 30% of adults are aged 65 or older [45], for single-person households, 70% of the individuals are assumed to be adults (aged 19–64) and 30% seniors (aged 65+). In two-person households, we follow the same distribution for households without children (with 70% of these consisting of adult couples and 30% senior couples) while the remaining 30% are modeled as single-parent families with one adult and one child. Three-person households are assumed to consist of two adults and one child, four-person households of two adults and two children, and households with five persons include two adults, two children, and one senior.

### 4.2. Location Capacities and Variation

Except for homes, we distinguish four location types: (i) schools, (ii) workplaces, (iii) supermarkets, and (iv) social events. Each location instance is associated with a predefined *capacity* (see Table 2), and agents are evenly distributed among these instances by the MEmilio framework. Including the home household, each agent is permanently assigned to a specific instance of each location type, ensuring designated supermarkets, social event venues, and either a workplace or school for every agent.

As empirical reference networks for the optimization process described in Section 3.2, we used the well-known Office network [12] for workplaces, a Primary School network [11] for schools, the Supermarket network [13], and the Science Gallery network [14] for social events. In our previous work [24], we determined that the STEPS model [42] was the most consistent choice for most networks. Accordingly, we used STEPS as the underlying HuMM for all networks except the Science Gallery, which was not analyzed in our earlier study. In the current experiments, the Science Gallery network was best captured by the STEPS+RWP approach. Further details on the HuMM properties can be found in [24]. If more detailed information on location capacities, the number of location instances, or specific types of locations is available, it can easily be integrated into the simulation framework. The following paragraphs describe the setup of these locations, including selected capacities, visitation rates, and other relevant properties.

#### 4.2.1. Schools and Workplaces

At the beginning of the simulation, each agent randomly selects a start time within a specified range, which remains fixed throughout the simulation. The start time determines when the agent attends school or work. All agents aged 19–64 are assigned to workplaces, those aged 5–18 attend schools. Each agent spends 9 h at work or 7 h at school. Agents do not attend schools or workplaces on weekends. The capacities of schools and workplaces are drawn from normal distributions, N(μ,0.3μ), with predefined minimum (10 for workplaces, 50 for schools) and maximum capacities (200 for workplaces, 500 for schools). The exact mean capacity, μ, varies depending on the scenario (Table 2).

#### 4.2.2. Supermarkets and Social Events

All individuals except those aged 0–4 may visit supermarkets or social events. We assume an average visitation rate of 0.5 per day for supermarkets and social events, meaning that each individual visits these locations on average every other day. Supermarket and social event locations are modeled with *fixed* capacities chosen via preliminary simulations. Specifically, we observed peak visitor loads and set each location’s capacity so that it is only rarely exceeded. Unlike home, work, or school, supermarket and social event visits occur irregularly due to the probabilistic visitation rate and the stochastic nature of the process. For example, in the “small” configuration (see Section 4.3), we include three supermarkets, each of capacity 35, and three social event venues, each of capacity 60. At any hour, if a location’s capacity is reached, new visitors are redirected home. Conversely, if fewer agents are present than the capacity, the corresponding nodes in the HuMM-based contact network are temporarily removed, ensuring each location’s contact network matches the actual number of occupants. To better reflect behavioral differences between weekdays and weekends, agents must spend a minimum of four hours at social events during weekends (compared to two hours on weekdays) before returning home. It is important to note that agents only visit supermarkets and social events when they are not occupied with work or school duties; this constraint is incorporated into the visitation rate so that the average remains consistent regardless of the number of free hours in an agent’s daily schedule.

#### 4.2.3. Households

Households are treated as fully connected networks since no reliable data were available to the authors to model the households’ internal contact patterns with our approach. For the purpose of demonstrating the potential of our method, it is reasonable to assume strong interactions among household members. The household sizes are determined by the composition described in Section 4.1 so that our simulation incorporates one-, two-, three-, four-, and five-person households. In this context, the household serves as the default location for an agent.

### 4.3. Experimental Setup

We investigate two complementary sets of scenarios to explore the potential of our approach for generating more realistic, high-resolution temporal contact networks.

#### 4.3.1. Scaling Population Size

In our first set of experiments, we keep the location capacities but scale the population by simulating 1000, 2000, and 5000 households. We label these TCN1000, TCN2000, and TCN5000. When increasing households, we proportionally increase the number of schools, workplaces, supermarkets, and social event venues, but *not* their individual capacities. This allows us to isolate the effect of population size from the effect of location size. The fixed capacity distributions of all three networks correspond to the *medium* setting of Table 2, i.e., $\mu_{\mathrm{School}}=200$ and $\mu_{\mathrm{Work}}=50$.

#### 4.3.2. Scaling Location Capacity

In this experiment, we fix the total number of households at 1000 (“TCN1000”) and only vary the capacities. We label these variants *small*, *medium*, and *large*. Following the description of Section 4.2, we model the capacities of workplaces and schools by drawing from a normal distribution until the total capacity of each location type equals or exceeds the number of assigned agents, while supermarkets and social events have fixed capacities. All agents are evenly distributed among the available locations. Table 2 shows three capacity settings (with σ=0.3μ).

## 5. Temporal Contact Network Analysis

In this section, we analyze the key structural and temporal properties of the generated Temporal Contact Networks (TCNs). As described in Section 4.3, we investigate the impact of varying location capacities as well as varying household counts on network characteristics and epidemic properties.

### 5.1. Impact of Scaling Population Size

In the following, the results for our first set of experiments on altering the number of households are presented. Apart from basic network statistics, the temporal contact dynamics are investigated with a focus on epidemic spreading properties.

#### 5.1.1. Network Statistics

Table 3 provides a summary of the basic properties of the TCNs generated, all under the *medium* configuration as described in Section 4.3. The reported metrics include the total number of nodes, the average number of active nodes (i.e., nodes involved in at least one contact during a given temporal snapshot), the total number of edges, the average degree, and the maximum diameter of the aggregated network.

As the number of households increases from 1000 to 5000, both the total number of nodes (agents) and the number of edges in the network scale proportionally. Despite this significant increase in total agent count, the average degree remains relatively constant across the three network sizes. This consistency is expected because the individual capacities for schools, workplaces, supermarkets, and social events were kept fixed. As a result, each agent encounters a similar number of neighbors within each location type, regardless of the total population size. Additionally, the maximum network diameter consistently stays between 9 and 11, indicating that the overall network extent in each temporal snapshot is maintained even as the network scales up. The share of active nodes also remains stable at around 82% across all three network sizes.

#### 5.1.2. Daily Contact Patterns

Figure 3a displays the daily contact counts for the TCN1000, TCN2000, and TCN5000 networks. We observe a clear day–night cycle, with contacts peaking during the daytime and dropping at night. As expected from our parameterization, including visitation rates, there are fewer contacts on weekends compared to weekdays. This reduction is consistent with the simulation settings described in Section 4.2.1 and Section 4.2.2. The networks with larger populations show a higher total number of daily contacts, indicative of the higher number of agents involved. Additionally, the minimum number of contacts never reaches zero, which is due to the households being fully connected networks as explained in Section 4.2. The minimum contact value is therefore driven by the basic interaction among household members.

In Figure 3b, we present the normalized contact counts by dividing the total daily contacts by the number of agents in the network. The resulting curves for TCN1000, TCN2000, and TCN5000 are largely overlapping, indicating that the interaction patterns across networks of different sizes remain comparable when adjusted for population scale.

Figure 4 illustrates the temporal evolution of the average degree for active nodes (i.e., excluding isolates) in TCN1000, TCN2000, and TCN5000 over 48 h. Notably, the mean degree remains largely consistent across the three networks, with an average value of approximately 10 during daytime hours. The average degree in the hours after work—when agents are engaged in activities at supermarkets or social events—remains comparable to the nighttime values. This observation is primarily attributed to the fully connected household networks, which provide a constant baseline of contacts that moderates the overall degree even during periods of reduced external activity. The maximum degree values are similar across the networks, ranging between 30 and 40.

Figure 5a shows the distribution of contact durations for TCN1000, TCN2000, and TCN5000. All three networks display similar contact duration patterns, suggesting that changing the total number of households—and hence the total number of agents—does not substantially alter the fundamental contact dynamics.

Figure 5b illustrates the degree distributions. As is common in real-world contact networks, the degree distributions exhibit heavy-tailed characteristics (i.e., some individuals have many more contacts than others) [46]. The uniformly distributed points for lower degrees are a result of the fully connected home networks. Notably, these distributions closely overlap for TCN1000, TCN2000, and TCN5000, underscoring that our approach maintains stable local interaction patterns even under population scaling.

#### 5.1.3. Epidemic Spreading

To illustrate how infections spread in the generated TCNs, we conducted a series of SIR simulations. In our second set of experiments, in Section 5.2, we identified that a β/γ ratio of 0.6—i.e., β=0.006 and γ=0.01—yielded the largest difference in the final number of infections. Therefore, to assess whether the extent of infection varies with population size, we employed these parameter settings. We randomly initialized 5% of the nodes as infected and performed 1000 simulation runs to ensure statistically robust results.

Figure 6 shows the absolute (Figure 6a) and normalized (Figure 6b) number of infected agents over time for TCN1000, TCN2000, and TCN5000.

The timing of epidemic peaks is broadly similar across the three networks, reflecting the same day–night contact pattern. However, the absolute number of infected individuals increases noticeably in larger networks.

Figure 6b demonstrates that, when normalized by the total number of nodes, the fraction of infected individuals remains largely similar across different population sizes. This invariance reinforces the claim that our temporal network generation approach produces consistent underlying contact patterns and temporal structures, independent of the overall network size generated.

In addition to examining the temporal infection curves for a single β/γ ratio (Figure 6), we also explore how the cumulative fraction of infections varies when changing the ratio β/γ. Figure 7 shows that for low β/γ, the epidemic generally remains small, whereas for higher values, a substantial fraction of the population becomes infected. Notably, TCN1000, TCN2000, and TCN5000 yield nearly identical outcomes across all β/γ ratios tested, confirming that the generated temporal contact networks exhibit size-invariant epidemic behavior.

Overall, the similar trajectories of the normalized number of infections across different population sizes indicate that the underlying contact patterns and resulting network dynamics are consistent. As expected, the absolute number of infections increases with population size due to the higher number of initially infected individuals. Nevertheless, the spreading speed and extent remain comparable when normalized.

In the second series of experiments, we vary the capacities of locations, demonstrating how changes in location characteristics can influence the resulting network dynamics.

### 5.2. Impact of Scaling Location Capacity

In our second set of experiments, we fix the total population at 1000 households but vary the capacities and number of instances of schools, workplaces, and social venues (as explained in Section 4.3). Unlike the population-scaling experiments in Section 5.1, where the overall network structure remained relatively constant, modifying location sizes here alters the connectivity more substantially. Additionally, we investigate network metrics of the resulting location sub-graphs and discuss differences.

#### 5.2.1. Network Statistics

Table 4 reports time-averaged metrics for the three TCN1000 variants, small, medium, and large. In contrast to scaling the population, increasing location capacities in TCN1000-large leads to a noticeable increase in average degree compared to TCN1000-small and TCN1000-medium. Specifically, the time-averaged degree grows from approximately 3.32 in the small configuration to 4.00 in the medium setup, and further increases to 4.22 in the large configuration.

The same trend holds for the *active* degree, reflecting that larger-capacity schools, workplaces, and social venues allow more agents to be present simultaneously, thus forming denser sub-networks.

Similarly, the maximum diameter grows from 9 hops in both the small and medium configurations to 13 in the large configuration, while the median diameter increases from 4 over 5 to 7.

This indicates that even though local cliques become denser when more agents gather in a single location, the global reach of connections also expands, creating longer shortest paths across different subgroups.

To further assess the temporal characteristics of the generated networks, we evaluated the burstiness coefficient *B* based on inter-contact times, following established approaches [47,48]. Burstiness captures the irregularity of interaction patterns, where frequent bursts of contacts alternate with long inactive periods, reflecting the non-Poissonian nature of human behavior. For TCN1000-small, the average burstiness was 0.312 (median 0.293), with 18.7% of nodes exhibiting B>0.5. TCN1000-medium showed a slightly higher average burstiness of 0.361 (median 0.353) and 19.4% of nodes above the threshold, while TCN1000-large resulted in an average of 0.371 (median 0.362) and 19.1% of nodes with strong burstiness. These values indicate moderately bursty dynamics across all configurations, with about one-fifth of the agents exhibiting pronounced burstiness, as expected in structured but temporally variable contact settings.

#### 5.2.2. Daily Contact Patterns

Figure 8 contrasts the absolute and normalized number of daily contacts over a 10-day period. Smaller but more numerous venues in TCN1000-small yield fewer overall edges, whereas larger venues in TCN1000-large produce denser interactions throughout the day. Figure 8a,b show almost identical curves, underscoring that a change in the underlying network structure has occurred.

Figure 9 illustrates the 48-h progression of the average active degree for the small, medium, and large configurations. TCN1000-small shows the lowest active degree of approx. 8 during daytime hours. TCN1000-large exhibits the highest average active degrees at around 10. These results support the table metrics, confirming that larger-capacity locations lead to an increase in average contacts. However, it remains difficult to quantify precisely how much larger this increase is, as it depends on multiple factors such as the properties of the underlying parameterized human mobility models and the specific capacity distributions of the location types. Further investigations are needed to better characterize the relationship of location capacities and average contacts.

Overall, the capacity-scaling results demonstrate that adjusting location sizes, rather than the population itself, has a pronounced effect on the connectivity and structure of the generated TCNs, thereby providing a complementary perspective to the population-scaling experiments in Section 5.1. This aligns with our expectation that a well-designed simulation framework should exhibit invariance under population scaling, while structural changes—such as those introduced by varying location capacities—should meaningfully affect the resulting temporal contact network and, consequently, influence epidemic spreading dynamics.

Figure 10 compares the distributions of contact duration (panel (a)) and node degrees (panel (b)) for TCN1000-small, TCN1000-medium, and TCN1000-large. In contrast to the pronounced changes seen in daily contact counts and average degrees, the *contact duration distribution* itself (panel (a)) remains largely unchanged across all three capacity settings. Only a slight increase in average contact duration is observed for TCN1000-small. This can be attributed to the nature of the simulated locations: in smaller-capacity environments, agents are more likely to repeatedly encounter the same individuals, given that the underlying human mobility parameters remain unchanged. As a result, while each agent maintains a similar number of total interactions, these interactions are distributed among a smaller and less diverse set of contacts. When aggregating contact durations over an hour, this leads to longer average durations per contact, but a lower degree, as agents connect with fewer unique individuals. This effect is also reflected in the degree distribution (panel (b)), where TCN1000-small shows slightly lower degrees, which is consistent with the more fragmented gatherings resulting from smaller (yet more numerous) venues. Despite these observable differences, the overall heavy-tailed degree pattern persists across all three networks, consistent with typical real-world contact dynamics.

#### 5.2.3. Epidemic Spreading

We next evaluate SIR epidemic dynamics to measure how location-capacity scaling influences infection propagation in the network. Figure 11 shows the absolute (panel (a)) and normalized (panel (b)) number of infected individuals over time for the three TCN1000 variants under β=0.006 and γ=0.01. Consistent with the previous results, TCN1000-large exhibits the highest absolute infection counts, while TCN1000-small remains comparatively lower. However, the normalized infection curves mirror the absolute ones, indicating that changes in location capacity do affect the underlying network structure and thus the overall spread of infection.

Finally, Figure 12 plots the cumulative fraction of infected individuals as a function of β/γ. Here, we observe notably fewer infections in TCN1000-small than in the larger-capacity networks for intermediate β/γ values, roughly between 0.4 and 1. The largest gap of about 0.2 occurs near β/γ≈0.6. For lower ratios around β/γ=0.2, the epidemic quickly dies out in all scenarios, yielding similarly small final sizes; for higher ratios above 2, rapid spread overwhelms the network regardless of capacity differences, causing the final epidemic sizes to converge.

This experiment further characterizes our temporal network generation approach by demonstrating how varying the available spaces and their capacities modifies the underlying contact dynamics and, consequently, the propagation of infectious diseases. Importantly, these results also underscore the *size-invariant* nature of our generated networks: while location size affects local contact intensity and overall connectivity, the framework exhibits consistency in key structural properties across different population scales (Section 5.1) and capacity settings.

As described in Section 4, the framework is highly parameterizable, allowing integration of diverse household data, location properties, and specific venue-level network assumptions. Although the exact realism of the simulated contacts depends on the availability and quality of the input parameters, our findings show that even under basic modeling assumptions, the generated temporal contact networks capture several key structural patterns observed in real-world settings. These include a characteristic day–night cycle in contact activity, heavy-tailed degree and contact duration distributions, and an overall size-invariant structure that persists across varying population sizes. Hence, the model is well-suited for diverse applications in epidemic analysis, including risk assessment, digital contact tracing simulations, and broader investigations into network-based disease propagation.

### 5.3. Location Sub-Graphs

In the following, the location-specific sub-graphs that form at schools, workplaces, supermarkets, social events, and households are analyzed. By focusing on active nodes at each location at a given time step, we obtain a series of smaller temporal sub-graphs.

Figure 13 compares the time evolution of two standard network metrics—average betweenness centrality and average degree—for each location type over the first 48 h of the simulation. For this analysis, we focus only on sub-graphs resulting from TCN1000-medium.

As shown in Figure 13a, supermarket sub-graphs exhibit the highest average peak in betweenness as well as the highest maximum value. This is intuitive, as supermarkets tend to experience high fluctuations in occupancy and serve as bridges between otherwise disjoint groups of agents. The social event sub-graphs also display relatively high betweenness values, though with slightly less pronounced peaks than supermarkets. In contrast, the school and workplace sub-graphs have lower average betweenness scores; in these environments, agents experience fewer fluctuations, forming dense clusters with multiple parallel potential pathways that reduce individual node centrality. Finally, the household (or home) sub-graphs—being fully connected cliques—produce near-zero betweenness, as every pair of household members is directly linked.

Figure 13b indicates that the average degree at peak times is comparable across the school, workplace, and supermarket sub-graphs. The social event sub-graphs, however, tend to have a slightly lower average degree. As expected, the household sub-graphs maintain consistently low average degree values due to their small, fully connected nature.

Overall, these findings confirm that location-specific attributes—such as fluctuations in attendance and inherent clustering—play a significant role in shaping the structure of contact networks. The high betweenness in supermarkets and social events is a consequence of their less clustered, more variable interaction patterns, while the denser structures at schools and workplaces yield lower betweenness. Households, by design, remain closely connected with both low degree and betweenness.

## 6. Conclusions

The results presented in this paper underscore the value of integrating Bayesian-optimized human mobility models into an agent-based epidemic framework. A key strength of our approach is the stability of the network structure across varying population sizes, reflecting the size-invariant properties of the generated temporal contact networks. This consistency arises because venue capacities remain fixed and agents experience similarly distributed contact patterns regardless of the total population. Epidemiologically, such scale invariance is highly advantageous, as it enables researchers to extrapolate insights from smaller simulations to larger populations with minimal distortion of network dynamics. Furthermore, the insights gained from independently varying population sizes and location capacities now provide a solid foundation for exploring their combined effects on contact dynamics and epidemic outcomes.

Another major advantage of our approach is its ability to produce sufficiently resolved temporal contact data, essential for individual-level analyses such as Digital Contact Tracing. Compared to more aggregated models, our approach permits the fine-grained modeling of contact events in heterogeneous settings. This granular perspective is often overlooked in previous work, yet it is crucial for real-world interventions that focus on tracking and isolating potentially infectious individuals. By coupling census-based demographic data with mobility-derived contact structures, our framework also enables researchers to explore interventions targeted at specific population groups or location types, while still preserving the fundamental characteristics of realistic human mobility.

Despite these strengths, there is scope for further refinement. A prominent limitation in our current implementation is the simplistic treatment of households as fully connected sub-networks. Empirical data on within-household contact structures can be difficult to obtain, and the degree of heterogeneity in family composition, shared flats, or communal living situations (e.g., assisted living or dormitories) is significant. Future research could develop more nuanced models of household contact patterns. Additionally, we used a single empirical reference network for each location type; future implementations could expand the range of reference datasets to capture more diverse types of locations, e.g., different categories of social venues. Even though the stochastic processes in our simulations do introduce variability across multiple instances of the same location type, providing multiple baseline networks for each category would further improve the fidelity of the generated contact structures.

In summary, our framework demonstrates the feasibility and value of embedding high-resolution, data-driven mobility models into epidemic simulations, enhancing both the realism and policy relevance of epidemic forecasts. Further improvements in household modeling and the diversity of location-specific reference data represent promising next steps for advancing this approach.

## Figures and Tables

**Figure 1 entropy-27-00507-f001:**
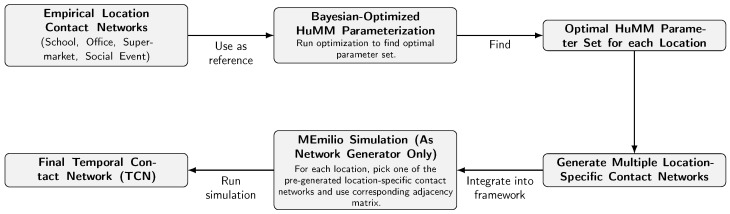
From empirical reference networks to final TCNs.

**Figure 2 entropy-27-00507-f002:**
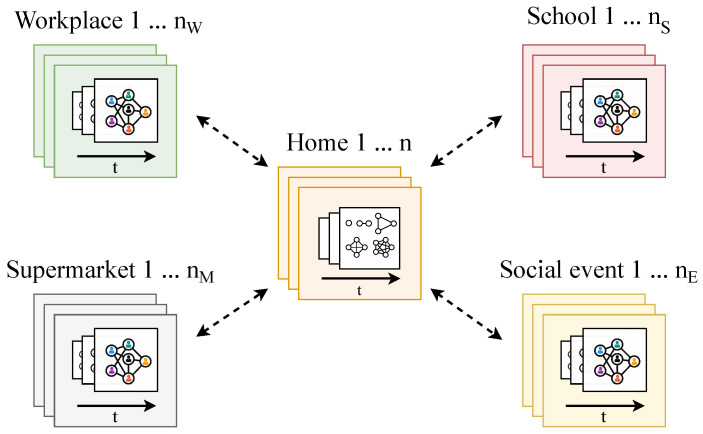
Schematic overview of simulation. The diagram displays multiple instances (from 1 to *n*) of environments—schools, workplaces, homes, supermarkets, and social events—each assigned a pre-generated, location-specific contact network.

**Figure 3 entropy-27-00507-f003:**
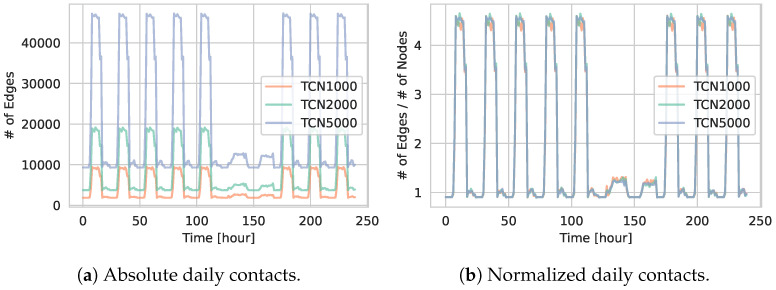
(**a**) Temporal evolution of the *absolute* number of contacts (# of edges) over 10 days for TCN1000, TCN2000, and TCN5000. (**b**) Same metric but *normalized* by the total number of agents in each network.

**Figure 4 entropy-27-00507-f004:**
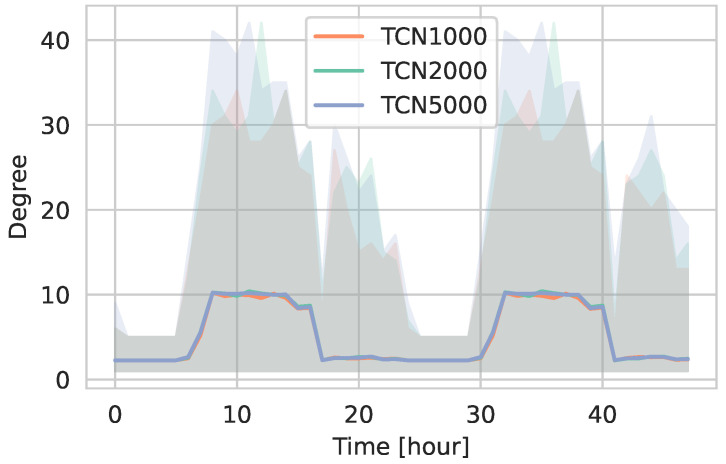
Temporal evolution of the degree over 48 h for active nodes in TCN1000, TCN2000, and TCN5000. The solid line represents the mean degree, the shaded areas indicate the minimum and maximum values.

**Figure 5 entropy-27-00507-f005:**
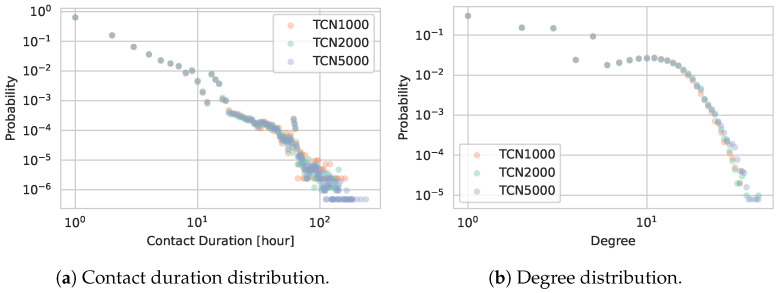
Comparison of contact patterns for TCN1000, TCN2000, and TCN5000.

**Figure 6 entropy-27-00507-f006:**
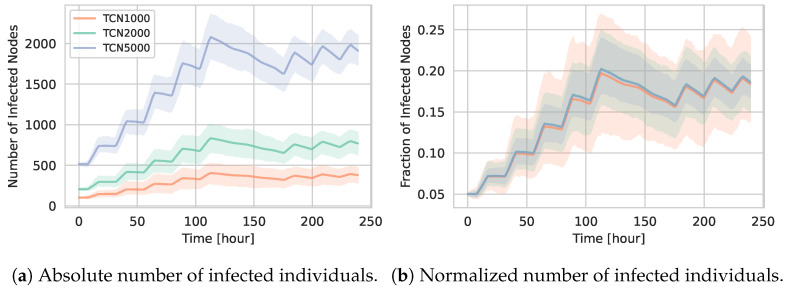
Epidemic progression on temporal contact networks (TCNs) for different population sizes under the medium configuration. Panel (**a**) shows the absolute number of infected individuals over time, while panel (**b**) presents the corresponding values normalized by the total number of agents.

**Figure 7 entropy-27-00507-f007:**
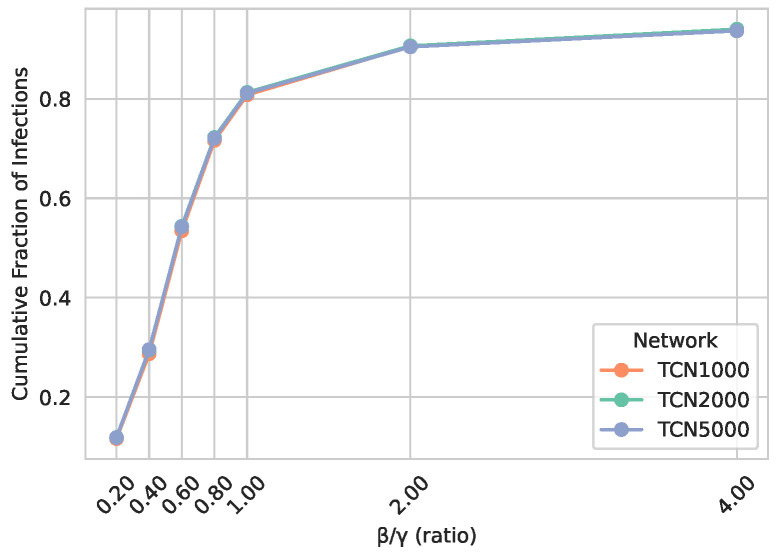
Cumulative fraction of infections of TCN1000, TCN2000, and TCN5000 across various β/γ ratios. γ is fixed at 0.01, and β is varied accordingly to yield β/γ values from 0.2 to 4.

**Figure 8 entropy-27-00507-f008:**
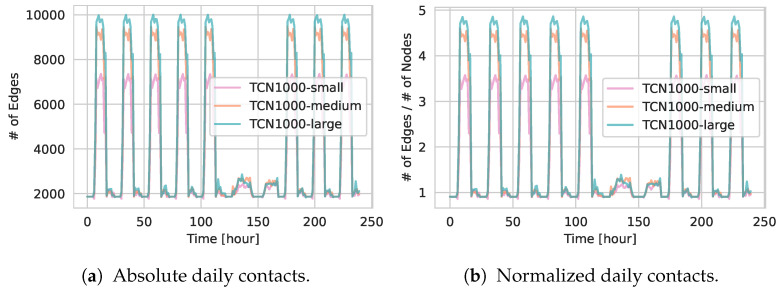
(**a**) Temporal evolution of the *absolute* number of contacts over 10 days for TCN1000-small, TCN1000-medium, and TCN1000-large. (**b**) Corresponding values normalized by the total number of agents in each network.

**Figure 9 entropy-27-00507-f009:**
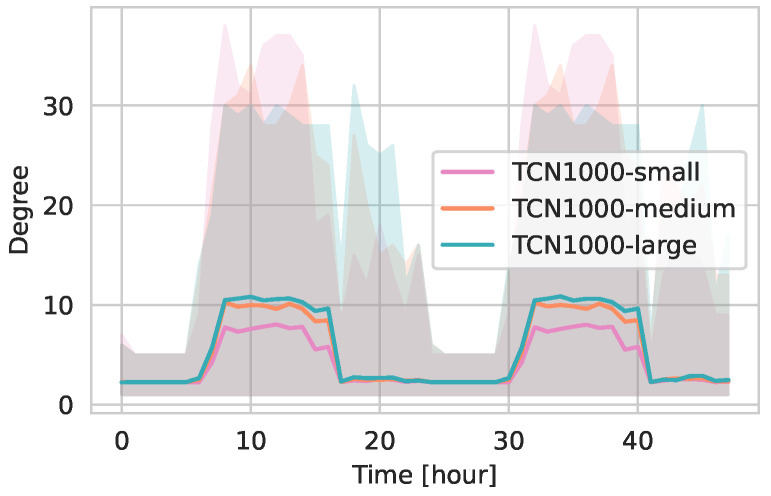
Temporal evolution of the degree (active nodes) over 48 h for TCN1000-small, TCN1000-medium, and TCN1000-large. The solid line represents the mean degree, the shaded areas indicate the minimum and maximum values.

**Figure 10 entropy-27-00507-f010:**
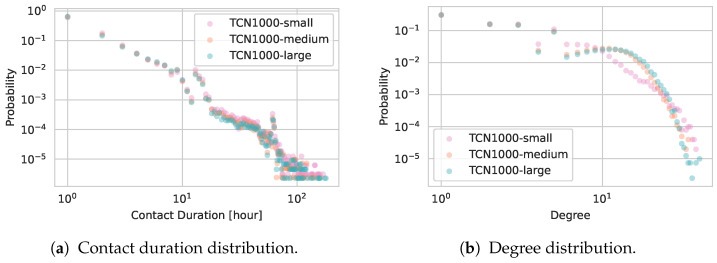
Comparison of (**a**) contact duration and (**b**) degree distributions for TCN1000-small, TCN1000-medium, and TCN1000-large. While contact duration patterns remain nearly identical, the degree distribution in the small-capacity scenario skews slightly lower.

**Figure 11 entropy-27-00507-f011:**
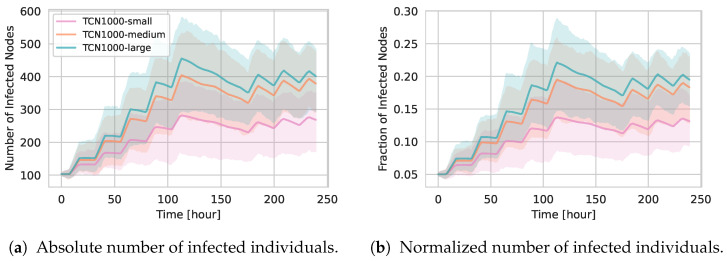
Epidemic progression in TCN1000-small, TCN1000-medium, and TCN1000-large under β=0.006 and γ=0.01. (**a**) Absolute infections, (**b**) normalized by the total number of agents.

**Figure 12 entropy-27-00507-f012:**
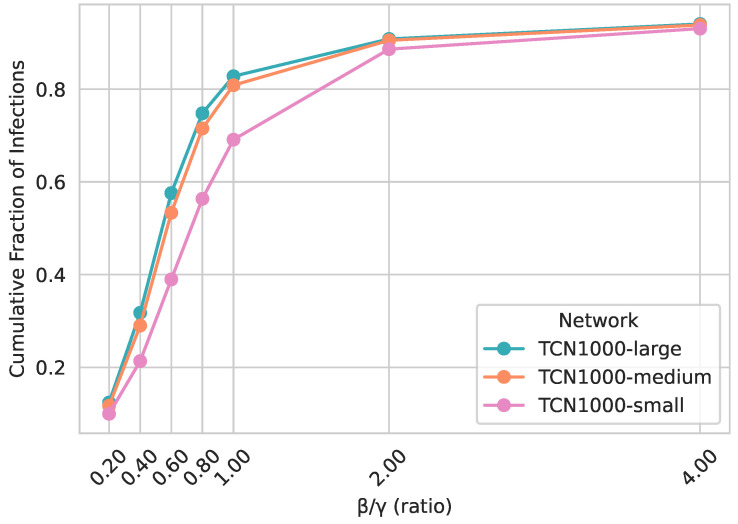
Cumulative fraction of infections in TCN1000-small, TCN1000-medium, and TCN1000-large across various β/γ ratios. γ is fixed at 0.01, and β is varied accordingly to yield β/γ values from 0.2 to 4.

**Figure 13 entropy-27-00507-f013:**
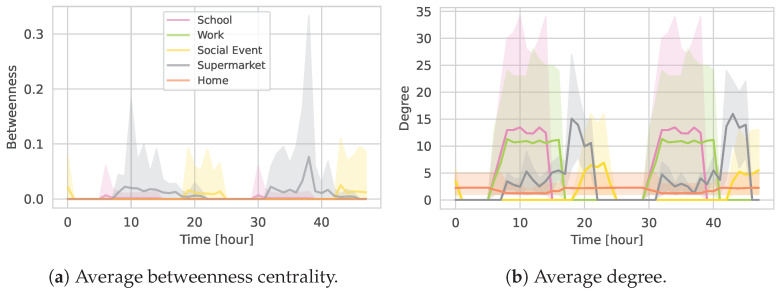
Temporal evolution of location-type sub-graphs in TCN1000 over the first 48 h. Panel (**a**) shows the average betweenness centrality, and panel (**b**) shows the average degree for nodes within each location type.

**Table 1 entropy-27-00507-t001:** Illustrative household composition and age assignments (rounded).

Household Type	Distribution
1-person (41%)	70%: 1 adult, 30%: 1 senior
2-person (33.5%)	70%: adult or senior couples,
	30%: 1 adult, 1 child
3-person (12%)	100%: 2 adults, 1 child
4-person (9.5%)	100%: 2 adults, 2 children
5+-person (4%)	100%: 2 adults, 2 children, 1 senior

**Table 2 entropy-27-00507-t002:** Mean capacities for schools and workplaces and corresponding capacities for supermarkets and social events in the TCN1000 scenarios. For supermarket and social event locations, the number of instances and the capacities of these instances are provided (e.g., 3 × 35).

Scenario	$\mu_{\mathrm{School}}$	$\mu_{\mathrm{Work}}$	Supermarkets	Social Events
TCN1000-small	100	20	3 × 35	3 × 60
TCN1000-medium	200	50	2 × 50	2 × 90
TCN1000-large	400	100	1 × 100	1 × 180

**Table 3 entropy-27-00507-t003:** Basic Network Metrics for TCN1000, TCN2000, and TCN5000 (all under the medium configuration as explained in Section 4.3). *Active nodes* are those that have contacts in the respective temporal snapshot. The *Avg Degree* column reports the degree averaged over all nodes, with the active node average provided in parentheses.

Network	# Nodes	Avg # Active Nodes	# Edges	Avg Degree (Overall/Active)	Diameter (Max/Median)
TCN1000	2058	1690.15 (82.13%)	987,777	4.00/4.71	9/5
TCN2000	4118	3379.36 (82.04%)	2,001,219	4.05/4.78	11/5
TCN5000	10,284	8456.24 (82.22%)	4,997,222	4.05/4.76	10/5

**Table 4 entropy-27-00507-t004:** Basic Network Metrics for TCN1000-small, TCN1000-medium, and TCN1000-large.

Network	# Nodes	Avg # Active Nodes	# Edges	Avg Degree (Overall/Active)	Diameter (Max/Median)
TCN1000-small	2057	1688	818,351	3.32/3.93	9/4
TCN1000-medium	2058	1690	987,777	4.00/4.71	9/5
TCN1000-large	2057	1690	1,042,295	4.22/4.97	13/7

## Data Availability

The empirical reference network datasets used in this study are freely available and can be accessed at the following repository: http://www.sociopatterns.org/datasets/ (accessed on 24 March 2025). The supermarket data used in this study are available upon request due to legal restrictions. All temporal contact network datasets generated in this work are provided at https://doi.org/10.5281/zenodo.15076221 (accessed on 24 March 2025).

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
