# Peer review of "Integrating Human Mobility Models with Epidemic Modeling: A Framework for Generating Synthetic Temporal Contact Networks"

_entropy, 2025, doi:10.3390/e27050507_

Round 1
Reviewer 1 Report
Comments and Suggestions for Authors
This manuscript presents an innovative framework for generating high-resolution temporal contact networks by integrating Bayesian-optimized human mobility models (HuMMs) with the MEmilio epidemic simulation framework. The manuscript addresses a critical gap in epidemic modeling by providing scalable, realistic contact networks that capture interactions across diverse environments. The methodological rigor and comprehensive evaluation of network properties and epidemic dynamics are commendable. Below are specific areas for improvement to enhance the clarity and impact of your work:
- The study relies on a single empirical network per location type (e.g., one office network for workplaces), which may not capture variability within categories (e.g., different workplace sizes or cultures). Expanding the reference dataset to include multiple networks per location type, or discussing how stochastic variability in HuMMs mitigates this limitation, would strengthen the analysis.
- While the paper mentions aggregation to hourly resolution, it does not explore the implications of finer (e.g., minute-level) or coarser resolutions. Including a brief analysis of how temporal granularity affects epidemic dynamics or network properties, or justifying the choice of hourly resolution, would provide more complete methodological context.
- The loss function weights (e.g., 5 for ΔImax, 3 for ΔTImax) are noted as determined empirically but lack theoretical or comparative justification. Providing a rationale for these weights or referencing sensitivity analyses would help demonstrate their robustness.
- Some minor revisions would improve the manuscript: ensuring consistency in figure/table references (e.g., "panel a" should be "panel (a)" in Line 520), and clarifying terms like "scale-free" in the context of contact networks (Line 636), as scale-free properties are not explicitly measured.
This manuscript makes a substantial contribution to epidemic modeling by introducing a scalable, data-driven framework for generating synthetic temporal contact networks. Addressing the suggested improvements—particularly regarding household modeling, reference network diversity, and computational considerations—will further strengthen the work. The study's innovative methodology and rigorous evaluation position it as a valuable resource for researchers and policymakers alike.
I recommend acceptance after minor revisions.
Author Response
Comment 1:
The study relies on a single empirical network per location type (e.g., one office network for workplaces), which may not capture variability within categories (e.g., different workplace sizes or cultures). Expanding the reference dataset to include multiple networks per location type, or discussing how stochastic variability in HuMMs mitigates this limitation, would strengthen the analysis.
Response 1:
We extended the section Stochastic Repetitions and Final Selection (Section 3.2.3) to explicitly highlight the stochastic nature of both HuMM generation and SIR simulations. We clarified that multiple realizations are generated from different seeds and that the optimizer minimizes the mean loss across this ensemble. Additionally, we emphasized that while this mitigates overfitting to a single dataset, the generalizability of the model would benefit from incorporating a broader range of empirical reference networks.
Page 7, L 250 – 267
Comment 2:
While the paper mentions aggregation to hourly resolution, it does not explore the implications of finer (e.g., minute-level) or coarser resolutions. Including a brief analysis of how temporal granularity affects epidemic dynamics or network properties, or justifying the choice of hourly resolution, would provide more complete methodological context.
Response 2:
We extended the relevant paragraph to clarify our choice of hourly granularity. We now explain that this resolution strikes a balance between behavioral realism and computational feasibility. Hourly aggregation aligns well with the typical timescale of meaningful behavioral changes in confined spaces (e.g., arrival peaks), while remaining sufficiently fine-grained for capturing epidemic dynamics. We acknowledge that future work could explore higher-resolution variants, and briefly mention the potential impact of finer/coarser resolution.
Page 8, L 292 – 302
Comment 3:
The loss function weights (e.g., 5 for ΔImax, 3 for ΔTImax) are noted as determined empirically but lack theoretical or comparative justification. Providing a rationale for these weights or referencing sensitivity analyses would help demonstrate their robustness.
Response 3:
Thank you for highlighting this point. We did not perform a formal sensitivity analysis, but instead adopted the same weighting strategy from our prior work—where preliminary tuning established these settings as robust. Specifically, we assigned the largest weights to SIR properties to ensure accurate reproduction of epidemic peaks, followed by edge counts and contact durations to preserve key network structural features. This configuration consistently yielded the best trade-off between epidemic-curve fidelity and structural realism. We adapted the explanation in Section 3.2.1. Bayesian Optimization Strategy.
Page 6, L 214 – 217
Comment 4:
Some minor revisions would improve the manuscript: ensuring consistency in figure/table references (e.g., "panel a" should be "panel (a)" in Line 520), and clarifying terms like "scale-free" in the context of contact networks (Line 636), as scale-free properties are not explicitly measured.
Response 4:
Thank you for the suggestions. We revised figures for consistency. Additionally, we clarified the terminology around "scale-free" by adopting the term "size-invariant" throughout the manuscript to avoid confusion.
Page 18, L 593 and L 604 / Page 19, L 640
Reviewer 2 Report
Comments and Suggestions for Authors
This paper presents an integrated approach for generating highly resolved temporal contact networks by coupling Bayesian-optimized human mobility models and validates their realistic interaction structures and infection dynamics. The model demonstrates that variations in population size do not affect the underlying network properties, highlighting its scalability. Overall, this study supports the simulation of epidemic propagation on synthetic temporal contact networks, with promising applications in urban management and intervention strategies. However, several minor improvements are suggested to enhance the validity of models and improve readability of the paper.
1. In the Methodology section, instead of mainly descriptive paragraphs, more technical details about the MEmilio Simulation Framework should be included to help readers quickly grasp the core concepts and functionalities of the framework.
2. While the model captures several intuitive features of daily human behavior, its effectiveness in representing the empirical characteristics of temporal contact networks should be discussed. In particular, attention should be given to the burstiness of human behavior, such as the power-law distribution of time gaps between consecutive mobility events (see Modeling temporal networks with bursty activity patterns of nodes and links, Physical Review Research, 2020).
3. Human mobility patterns differ significantly between weekdays and weekends. This discrepancy should be incorporated into the simulation settings for locations such as schools, workplaces, and supermarkets to improve the model’s realism.
4. How to interpret the large confidence interval of Figure 4 and Figure 9.
5. An overview of the spatial feature of mobility, particularly the origin-destination relationships between various locations—including long-distance travels that are critical to pandemic spread (see Strong Long Ties Facilitate Epidemic Containment on Mobility Networks, PANS Nexus, 2024)—should be discussed to demonstrate the model’s effectiveness in simulating collective emergent behaviors within an urban environment.
6. Several methods for obtaining human mobility data should be discussed in the Introduction to provide a comprehensive overview of the challenges involved in acquiring empirical personal contact data. Examples include Cellar Signal Data (CSD) as examined in Mobility in China, 2020: A Tale of Four Phases, National Science Review, 2021 and involving concerns raised in Society: Protect Privacy of Mobile Data, Nature, 2014. The statement that “networks aggregating contact patterns over longer periods, such as days, weeks, or even months, are especially useful for analyzing general trends and understanding population-level dynamics” would benefit from additional supporting references, such as The Antecedents and Consequences of Network Mobility, PNAS, 2023 and Heterogeneous Changes in Mobility in Response to the SARS-CoV-2 Omicron BA.2 Outbreak in Shanghai, PNAS, 2023.
Author Response
Comment 1:
In the Methodology section, instead of mainly descriptive paragraphs, more technical details about the MEmilio Simulation Framework should be included to help readers quickly grasp the core concepts and functionalities of the framework.
Response 1:
We thank the reviewer for this helpful suggestion. In response, we have expanded the description of the MEmilio framework in Section 3.1. The revised text now provides additional technical details, including the object-oriented design, core data structures (e.g., Person, Household, Location), and key implementation aspects such as the use of templated C++, parallelization with OpenMP and MPI, and general modeling capabilities like parameter spaces and mobility rules. We also explicitly state that in the present work, MEmilio is used solely as a generator for synthetic temporal contact networks, with all infection-related components deactivated.
Page 4, L 145 – 160
Comment 2:
While the model captures several intuitive features of daily human behavior, its effectiveness in representing the empirical characteristics of temporal contact networks should be discussed. In particular, attention should be given to the burstiness of human behavior, such as the power-law distribution of time gaps between consecutive mobility events (see Modeling temporal networks with bursty activity patterns of nodes and links, Physical Review Research, 2020).
Response 2:
We thank the reviewer for the suggestion. We have included an evaluation of the burstiness coefficient to more consistently assess the temporal properties of the generated networks. The results are now reported in Section 5.2.1 (Network Statistics), providing a quantitative view of the temporal variability across different configurations.
Page 16, L 524 – 534
Comment 3:
Human mobility patterns differ significantly between weekdays and weekends. This discrepancy should be incorporated into the simulation settings for locations such as schools, workplaces, and supermarkets to improve the model’s realism.
Response 3:
We thank the reviewer for the suggestion. Weekend behavior was already incorporated in the simulation design: agents did not attend schools or workplaces on weekends, and had longer minimum stays at social event locations compared to weekdays. To make this clearer, we have now explicitly stated these aspects in Section 4.2.1 (Schools and Workplaces) and Section 4.2.2 (Supermarkets and Social Events). Furthermore, we mention in Section 5.1.2 (Daily Contact Patterns) that weekend effects are reflected in the daily contact patterns, and have added a cross-reference to the relevant simulation settings for clarity.
Page 9/10, L 356 – 365 and 378 – 382
Comment 4:
How to interpret the large confidence interval of Figure 4 and Figure 9.
Response 4:
We would like to clarify that the shaded areas in Figure 4 and Figure 9 do not represent confidence intervals but instead show the minimum and maximum observed degrees over time, as described in the figure captions. The large range between minimum and maximum degrees is expected and reflects the heterogeneity of agent activity patterns in the simulation: for instance, elderly agents (who do not attend work or school) typically maintain very low degrees during daytime, while schoolchildren and working-age adults at schools and workplaces form much denser networks and thus exhibit much higher degrees.
Comment 5:
An overview of the spatial feature of mobility, particularly the origin-destination relationships between various locations—including long-distance travels that are critical to pandemic spread (see Strong Long Ties Facilitate Epidemic Containment on Mobility Networks, PANS Nexus, 2024)—should be discussed to demonstrate the model’s effectiveness in simulating collective emergent behaviors within an urban environment.
Response 5:
We thank the reviewer for raising this important point. In our current study, we focus on local contact dynamics within a small urban area or community comprising a few thousand households. At this scale, long-distance travel patterns (such as inter-city travel by train or bus) are not a primary driver of contact dynamics, and infections during transit are not explicitly modeled. Instead, our framework simulates daily movements between and especially within homes, schools, workplaces, supermarkets, and social venues, emphasizing short-range mobility and location-based contact structures. Nevertheless, we agree that incorporating infection events during longer trips (e.g., in public transport) is an important aspect for larger-scale or multi-regional modeling. Extending the MEmilio framework in this direction is an active area of development.
Comment 6:
Several methods for obtaining human mobility data should be discussed in the Introduction to provide a comprehensive overview of the challenges involved in acquiring empirical personal contact data. Examples include Cellular Signal Data (CSD) as examined in Mobility in China, 2020: A Tale of Four Phases, National Science Review, 2021, and involving concerns raised in Society: Protect Privacy of Mobile Data, Nature, 2014. The statement that “networks aggregating contact patterns over longer periods, such as days, weeks, or even months, are especially useful for analyzing general trends and understanding population-level dynamics” would benefit from additional supporting references, such as The Antecedents and Consequences of Network Mobility, PNAS, 2023, and Heterogeneous Changes in Mobility in Response to the SARS-CoV-2 Omicron BA.2 Outbreak in Shanghai, PNAS, 2023.
Response 6:
We thank the reviewer for this valuable suggestion. We have expanded the Introduction to provide a broader overview of data sources used in human mobility and contact studies. Specifically, we now discuss the use of Cellular Signaling Data (CSD) and cite the study "Mobility in China, 2020: A Tale of Four Phases" to illustrate large-scale mobility analysis based on mobile phone records. We also acknowledge the associated privacy concerns by referencing "Protect Privacy of Mobile Data". Furthermore, we added citations to "The Antecedents and Consequences of Network Mobility" and "Heterogeneous Changes in Mobility in Response to the SARS-CoV-2 Omicron BA.2 Outbreak in Shanghai" to support the relevance of aggregating contact patterns over longer periods for analyzing population-level dynamics.
Page 2, L 39 – 45
Round 2
Reviewer 2 Report
Comments and Suggestions for Authors
The authors have addressed all my concerns satisfactorily, and I recommend acceptance of the manuscript in its current form.